# Recent Advances in Zinc Oxide Nanoparticles (ZnO NPs) for Cancer Diagnosis, Target Drug Delivery, and Treatment

**DOI:** 10.3390/cancers13184570

**Published:** 2021-09-12

**Authors:** Sumaira Anjum, Mariam Hashim, Sara Asad Malik, Maha Khan, José M. Lorenzo, Bilal Haider Abbasi, Christophe Hano

**Affiliations:** 1Department of Biotechnology, Kinnaird College for Women, Jail Road, Lahore 54000, Pakistan; mariamhashim07@gmail.com (M.H.); gabby_sara.malik@outlook.com (S.A.M.); maha.daud10@gmail.com (M.K.); 2Centro Tecnológico de la Carne de Galicia, Avenida de Galicia 4, Parque Tecnológico de Galicia, 32900 San Cibrao das Viñas, Ourense, Spain; jmlorenzo@ceteca.net; 3Área de Tecnología de los Alimentos, Facultad de Ciencias de Ourense, Universidad de Vigo, 32004 Ourense, Spain; 4Department of Biotechnology, Quaid-i-Azam University, Islamabad 15320, Pakistan; bhabbasi@qau.edu.pk; 5Laboratoire de Biologie des Ligneux et des Grandes Cultures, INRAE USC1328, Eure & Loir Campus, University of Orleans, 28000 Chartres, France; hano@univ-orleans.fr

**Keywords:** zinc oxide nanoparticles, cancer, ROS, drug delivery, diagnosis, anticancerous activity

## Abstract

**Simple Summary:**

Despite breakthroughs in medicine, cancer remains one of the most feared diseases. Traditionally, chemotherapies have been the treatment of choice. However, concerns about stability and poor solubility prevent them from being widely used. Destruction of healthy cells, hair loss, and drug resistance are all common side effects. In this aspect, nanotechnology opens up new avenues for cancer treatment. ZnO NPs are one of the most valuable metal oxide nanoparticles in cancer treatment owing to their high biocompatibility and low toxicity. The ability of ZnO NPs to selectively trigger the formation of reactive oxygen species and induce apoptosis has been critically appraised in this review. They are also the best contenders for diagnosis and tailored drug delivery in cancer therapeutics. Here, we extensively summarize the role of ZnO NPs in cancer diagnosis, target drug delivery, and treatment, which will help in future research advancements in the field of cancer theranostic.

**Abstract:**

Cancer is regarded as one of the most deadly and mirthless diseases and it develops due to the uncontrolled proliferation of cells. To date, varieties of traditional medications and chemotherapies have been utilized to fight tumors. However, their immense drawbacks, such as reduced bioavailability, insufficient supply, and significant adverse effects, make their use limited. Nanotechnology has evolved rapidly in recent years and offers a wide spectrum of applications in the healthcare sectors. Nanoscale materials offer strong potential for curing cancer as they pose low risk and fewer complications. Several metal oxide NPs are being developed to diagnose or treat malignancies, but zinc oxide nanoparticles (ZnO NPs) have remarkably demonstrated their potential in the diagnosis and treatment of various types of cancers due to their biocompatibility, biodegradability, and unique physico-chemical attributes. ZnO NPs showed cancer cell specific toxicity via generation of reactive oxygen species and destruction of mitochondrial membrane potential, which leads to the activation of caspase cascades followed by apoptosis of cancerous cells. ZnO NPs have also been used as an effective carrier for targeted and sustained delivery of various plant bioactive and chemotherapeutic anticancerous drugs into tumor cells. In this review, at first we have discussed the role of ZnO NPs in diagnosis and bio-imaging of cancer cells. Secondly, we have extensively reviewed the capability of ZnO NPs as carriers of anticancerous drugs for targeted drug delivery into tumor cells, with a special focus on surface functionalization, drug-loading mechanism, and stimuli-responsive controlled release of drugs. Finally, we have critically discussed the anticancerous activity of ZnO NPs on different types of cancers along with their mode of actions. Furthermore, this review also highlights the limitations and future prospects of ZnO NPs in cancer theranostic.

## 1. Hallmarks of Cancer

Cancer is a primary cause of death and a significant impediment to extending life expectancy, with 10 million fatalities in 2020, according to a WHO report. The number of newly diagnosed cancer cases is projected to increase to 29.5 million per year, and projected cancer-related deaths are expected to increase to 16.4 million per year by 2040 [1,2]. Despite novel medicinal techniques and technological advancements, cancer remains one of the most lethal diseases [3,4]. It is a disease characterized by uncontrolled growth of the cells [5]. Cancer cells normally exist in the body and function like regular cells. Cancer is triggered when uncontrolled cell division leads to excessive cell proliferation. Cancer cells have the ability to build up their own blood supply, break away from the original organ, migrate, and spread to other body organs [6]. Proliferation can be lethal if allowed to persist and spread, and it is mainly due to the ability of cancer cells to ignore signals, whereas normal cells only replicate when they encounter signals. It may also be due to the mutations in genes offering checkpoints for cell proliferation [7]. Genomic instability may also lead to cancer, in which increased alterations occur in the genome during the entire life cycle of dividing cells [8]. Another phenomenon through which cancer progresses is angiogenesis, leading to the development of new blood vessels for proper supply of nutrients and oxygen to metastatic cancer cells [9]. Cancer is not caused due to a single factor. In fact, numerous internal and external factors are involved in the initiation of cancer. Internal factors include mutations in genes, genetic disorders, and hormone imbalances, and external factors include exposure to radiation, malnutrition, infectious agents, tobacco use, etc. [10]. Around 90% of cancer-related deaths are caused by tumor spread, which is referred to as metastasis [11]. A hereditary gene defect is responsible for only 5–10% of all cancers. Despite the fact that all tumors are caused by a combination of mutations [12,13], these mutations occur as a result of interactions with the environment and carcinogens [14,15]. The main hallmarks of cancer are shown in Figure 1.

## 2. Conventional Therapy vs. Nanotechnology-Based Therapy for Treatment of Cancer

In conventional therapy, surgery, chemotherapy, radiotherapy, immunotherapy, phototherapy, and hormonal therapy are only a few of the therapeutic options for cancer as demonstrated in Figure 2C. Surgery could not be the final decision for all types of cancers and, chemotherapy in which medications are delivered intravenously into the body, has been the most often used treatment for cancer patients until now [16]. Despite advancements in chemotherapy, radiation therapy, immunotherapy, phototherapy and hormonal therapy, the main concern with these treatments is their negative side effects, which destroy healthy cells of the body along with damaged cells, which makes them a poor choice [17]. Conventional therapies are also constrained due to their low solubility, inability to invade tumors, nonspecific targeting, and inducing significant damage to the immune system and other organs, hence offering a low survival rate [18]. Conventional therapies are also known to develop resistance after exposure to subsequent doses, as represented in Figure 2A. However, in some circumstances, for example, when tumors are detected early, cures can be achieved, but recurrence and resistance are prevalent. Therefore, designing a drug delivery system that exclusively delivers deadly chemicals to tumors while leaving healthy cells alone is a difficult task [19].

Nanoparticles (NPs) have attracted the attention of scientists in recent years because of their high efficacy and safety [20]. Therapeutic and diagnostic techniques based on nanotechnology have shown tremendous promise in improving cancer therapy in recent years [21,22]. Nanomedicine technologies have cleared the path for novel targeted cancer therapies by allowing therapeutic compounds to be encapsulated in nanoparticulate materials and delivered selectively to tumors via passive permeation and active internalization mechanisms. Employing NPs for therapeutic purposes has also been found to minimize resistance, addressing one of the most significant obstacles to conventional therapy as depicted in Figure 2B. Various studies are being conducted in order to identify more precise nanotechnology-based cancer treatments that have less adverse effects than conventional treatments [23]. Nanomedicine technology has progressed to the point that a number of nanomedicines for cancer treatment are already on the market, with many more in advanced phases of development and clinical testing employing various nanosystems, i.e., metallic nanoparticles, liposomes, quantum dots, carbon nanotubes, polymeric micelles, and nanospheres [19,24,25,26,27,28] as shown in Figure 2D.

## 3. Zinc Oxide Nanoparticles (ZnO NPs): Potential Candidate for Fighting Cancer

Among all NPs, zinc oxide nanoparticles (ZnO NPs) are one of the most exploited candidates in drug delivery, cancer diagnosis, and treatment due to their unique physical and chemical properties. ZnO NPs are not only used in fighting cancer, but also proved to be very efficient in fighting many other diseases and in a variety of other sectors such as cosmetics, electronics, and the textile industry as well [29,30,31,32]. ZnO NPs can be synthesized chemically, physically, or biologically. Precipitation, microemulsion, chemical reduction, sol-gel, and hydrothermal procedures are some of the few examples of chemical methods that consume a lot of energy and also require maintenance of high pressure or temperature during the synthesis process [33,34,35,36]. ZnO NPs can be manufactured via physical processes such as vapor deposition, plasma, and ultrasonic irradiation, which are less common than chemical methods [34,37,38]. Nonetheless, these procedures typically necessitate a high level of energy and heavy equipment, which raises the price of the items. Another method for obtaining ZnO NPs is by biological synthesis, which has emerged as a more environmentally friendly technology [39]. Irrespective of the method employed, all types of ZnO NPs have proved to be efficient in combating cancers, in terms of their diagnosis, treatment, and sustained/targeted release of anticancerous drugs [40,41,42].

ZnO NPs are one of the most widely used metal oxide NPs in a variety of sectors and research institutions since they possess significant applications [43]. Because of the small particle size of nano-ZnO, the human body can easily absorb zinc. Since ZnO NPs are relatively affordable and less toxic than other metal oxide NPs, they offer a wide range of other medicinal uses, including antimicrobial, anti-diabetic, anti-inflammatory, anti-aging and also in wound healing and bio-imaging [44,45,46,47]. ZnO NPs have a high biocompatibility, allowing it to be used in a therapeutic environment for antibacterial, antifungal, antiviral, and anticancer properties [48]. Several types of inorganic metal oxides, such as TiO_2_, CuO, and ZnO have been produced and have remained in current investigations, but ZnO NPs are the most interesting of these metal oxides since they are inexpensive to make, safe, and simple to prepare [49]. Furthermore, the following characteristics must be present in any agent intended for human consumption for treating various diseases. It should be nontoxic, must not react with food or the container, it should have a pleasant flavor or be tasteless, and it should not have an unpleasant odor. ZnO NPs are one such inorganic metal oxide that meets all of the aforementioned criteria, allowing them to be used safely as a medication, package preservative, and antibacterial agent [50,51]. Hence, the US Food and Drug Administration (FDA) has classified ZnO NPs as a “GRAS” (generally regarded as safe) substance [52]. ZnO NPs because, of their huge band gap (3.37 eV) and high exciton binding energy (60 meV), have a wide range of semiconducting capabilities, including strong catalytic activity, optics, UV filtering, anti-inflammatory, and wound healing [53]. They have also been widely used in cosmetics such as sunscreen lotions due to their UV filtering qualities [54]. ZnO NPs were first used in the rubber industry to provide wear resistance to rubber composites, improve the toughness and intensity of high polymers, and provide anti-aging properties [55,56]. ZnO NPs have also gained attention in biomedical imaging due to their ability to exhibit luminescence [57]. They have also attracted researchers towards the development of diagnostic tools, as they can also be employed in biosensing applications [58]. The unique attributes of ZnO NPs, due to which they can act as potent anticancerous agents, are listed in Figure 3.

ZnO NPs have also been studied as nanocarriers for a range of payloads, such as medicines, genes, proteins, and imaging agents [47,59]. Furthermore, ZnO NPs have also been exploited as a pH-sensitive nanocarrier for tumor-targeted medication administration and intracellular drug release because they dissolve easily at low pH [60,61]. Mounting evidence suggests that ZnO NPs are capable of destroying cancer cells by the production of reactive oxygen species (ROS), implying their potential as an anticancer agent [40,62]. Thus, these findings imply that ZnO NPs could be the best candidate for fighting cancer in terms of diagnosis, in vivo bio-imaging, sustained drug delivery, and as a multi-target anticancerous agent as illustrated in Figure 3.

## 4. Role of ZnO NPs in Diagnosis of Cancer

Despite advances in the development of novel therapy methods, cancer continues to be one of the main causes of death in humans. This is primarily due to the failure to detect malignancies at an early stage. Cancer may have progressed by the time it is discovered, making therapy difficult, if not impossible. To diagnose cancer at an early stage, it is vital to develop new diagnostic procedures or to improve the existing ones. To solve these bottlenecks, the recent progress in nanotechnology has made this possible. ZnO NPs are one of the best-suited nanomaterials to be used as sensitive tags for detecting various types of cancers due to their unique physiochemical features. Probes based on NPs paired with the relevant targeted molecules interact with biological systems and detect biological changes at the molecular level with remarkable precision [63].

Recent advances in unique properties of nanomaterials have heightened interest in their employment in biosensing sectors, resulting in substantial advancement in the fabrication of nanomaterial-based biosensors. The use of nanostructured materials to organize electrochemical sensing devices has a lot of appeal since they provide an electro-catalytic effect as well as a higher surface area, which is important for enzyme immobilization in electrochemical biosensors [64,65]. Early cancer detection, on the other hand, is crucial for patient survival and effective disease therapy, which demands delicate and specialized methods. Existing diagnostic methods (such as ELISA) are insufficiently sensitive, as they detect cancer biomarkers at later stages of the disease. Thus, approaches that are more realistic, quicker, and more affordable are needed over time [66,67]. Thus, the possibility of incorporating biosensors into point-of-care (POC) systems could be an attractive option. Biosensors for cancer detection have a wide range of applications.

Coupling electrochemical detection techniques with immunosensors and cyto-sensors allows for the development of quick, low-cost, and effective systems [67]. In a recent study, researchers have developed a screen-printed des-carboxy-prothrombin (DCP) immunosensor using ZnO NPs for accurate DCP assessment in the detection of liver cancer. DCP is a novel biomarker for detecting liver cancer that has a sensitivity of roughly 70% and a specificity of approximately 100% [68]. As a result, the DCP immunosensor developed is simple, cheap, and reliable, with the potential to be used at home to screen for early-stage liver cancer using a point-of-care approach [69]. Similarly, an enhanced Interdigitated Electrode (IDE) based nanobiosensor for detecting viral oncogenes was developed by coating gold doped ZnO nanorods on the top of IDE chips surface with viral DNA receptors of human papillomavirus subtype-16 (HPV-16). Due to the higher sensitivity and biocompatibility of the created nanohybrid film, HPV-16-E6 oncogene biosensors demonstrate excellent detection of HPV-16-E6 oncogene (cancer biomarker for HPV infected cervical cancers). By detecting viral E6 gene targets as little as 1fM, this sensor displayed extraordinary sensitivity. The sensor also has a durability of over 5 weeks, excellent repeatability, and great HPV-16 discriminating abilities. As a result, the proposed sensor is a versatile instrument with a lot of potential for clinical diagnosis, especially in economically challenged nations and areas [70].

Among different forms of vitamin B, choline is a crucial component in the human body, as it plays an important role in several metabolic pathways. Deviation from normal choline levels result in sickness. In patients with triple-negative breast cancer (TNBC), choline and its related chemicals are reported to be increased at higher levels. Therefore, detection of choline in cancer cells using a nano-interfaced electrochemical biosensor is also of great interest to researchers nowadays. Recently, a working electrode composed of glassy carbon with a ZnO NPs interface was developed for electrochemical detection. Drop casting was used to immobilize the choline oxidase (ChOx) enzyme on a nano-interface. The linear range of the developed biosensor was 0.3 mM to 5.1 mM, the detection limit (LOD) was 0.58 mM, and the quantification limit (LOQ) was 1.93 mM. The findings suggest that this biosensor could be utilized in clinical practice to identify breast cancer [71].

Nanopore layouts improve analytical performance by increasing the loading of active catalysts and allowing compounds to diffuse at faster rates [72]. Due to its cross-linked structure and ability to resist biodegradation, nanopore ZnO is considered to be the most stable [73]. Thus, manipulating the composition of materials during synthesis allows for the alterations in conductivity, distribution, shape, and size of ZnO nanofibers for their applications in developing biosensors. For the diagnosis of breast cancer, mesoporous ZnO nanofibers (ZnOnF) were synthesized using an electrospinning technique with the diameter in a range of 50–150 nm and anti-epidermal growth factor receptor 2 was used as a biomarker in conjugation. The outstanding impedimetric sensitivity of this immunosensor provides the fast detection (128 s) of cancer in a wide detection test range (1.0 fM–0.5 M). The suggested point-of-care cancer diagnostics have various advantages, including increased stability, quick monitoring, simplicity, cost effectiveness, and the capacity to identify other bio- and cancer markers [74].

ZnO-nanosheets were prepared using zinc nitrate and triethanolamine at neutral pH and low temperature. K562 cells, a model of leukemia, were treated with ZnO-nanosheets for 20 min before imaging. After incubation, the cells emitted a yellow-orange light, indicating that ZnO nanostructures had successfully penetrated the cells [75]. Similarly, a stable ZnO polymer with core shell NPs using the sol-gel method was synthesized, which exhibited photoluminescence in solution and high quantum yield. It was evaluated on human hepatoma cells using 3 nm ZnO-1 (green fluorescence) and ZnO-2 (yellow fluorescence), which did not show any toxicity against hepatoma cells. As these polymers are safe and cheap, they can be used in cell imaging as fluorescent probes in in vitro as well, implying potential uses in biological sectors [76].

## 5. ZnO NPs in Sustained and Targeted Delivery of Anticancerous Drugs

NPs are a good way to deliver site-specific drugs and bioactive agents in a controlled manner [77]. Drug formulation in biocompatible nanoforms is emphasized in pharmaceutical nanotechnology, which provides advantages in drug delivery. NPs improve drug efficiency and safety by improving bioavailability, providing targeted drug delivery, improving drug stability, and extending the drug’s impact on the target tissue [78]. ZnO is used in current drug delivery systems due to its ease of manufacture, low cost, customizable structure, non-toxicity, high drug-loading capacity, programmable drug release ability, and targeted delivery [79,80,81,82]. Porous ZnO structures such as porous nanotubes, porous nanobelt, porous nanorods, and porous cages have been successfully used in targeted drug delivery systems [83,84]. In order for the drug to be delivered efficiently, surface functionalization of NPs can be done, which is achieved using various agents, i.e., ligands, linker chains, drugs, and markers. Through specific molecular interactions such as receptor–ligand-based interactions [85,86], NPs accumulate in cells, which through endo/lysosomal escape on receiving an appropriate stimulus, release the drug, destroying its cognate target as depicted in Figure 4. Various types of internal and external stimulus are also involved in targeted delivery of anticancerous drugs. Thus, various ZnO nanostructures could be employed successfully in loading and targeted delivery of anticancerous drugs, as summarized in Table 1.

### 5.1. ZnO NPs: Carrier of Anticancerous Bioactive Compounds in Sustained Drug Delivery

Medicinal plants are a gift from nature to humanity, assisting them in their quest for improved health. Plants and their bioactive substances have been used in traditional medicine since the beginning of human history. Some medicinal plant species include phytochemicals that inhibit the progression and development of cancer by blocking cancer cell signaling pathways, activating DNA repair processes, and acting as antioxidants, demonstrating substantial anticancer action in terms of efficacy [87,88,89]. Among different plant bioactive compounds, taxifolin is considered a strong flavanol due to its unique structure with better antioxidant capabilities than other flavonoid compounds. In biomedical applications, however, its low bioavailability is a serious disadvantage. Biodegradable polymer coated ZnO NPs loaded with taxifolin were evaluated for their loading capacity and drug release behavior. The taxifolin-loaded ZnO NPs were found to be highly effective against the cancer cell line. Apoptotic signals were also observed to be induced by the produced particles in the selected cells. Even a lesser dose of 27 µg/mL was sufficient to inhibit MCF-7 cells. Further apoptotic experiments demonstrate that cell death is caused by fragmentation of the treated cells nuclear material. Thus, taxifolin-loaded ZnO NPs could be of prime importance in targeted drug delivery [90]. Similarly, functionalized nanohybrid hydrogels using L-histidine (HIS) conjugated chitosan with embedded ZnO NPs were developed for efficient delivery of polyphenol drugs such as naringenin (NRG), quercetin (QE), and curcumin (CUR). In the hybrid hydrogel, maximum loading efficiencies of 90.55%, 92.84%, and 89.89% were optimized for NRG, QE, and CUR, respectively. The hydrogel was stabilized by HIS–chitosan conjugation, which exhibited long-term drug administration at pH 5. When compared to free polyphenol drugs, the hybrid carrier showed a 15 to 30-fold increase in cytotoxicity in anticancer experiments on human skin carcinoma (A431) cells [91]. Likewise, ZnO NPs synthesized using the co-precipitation method were surface functionalized with PEG and beta cyclodextrin and loaded with a hydrophobic drug (curcumin). The brine shrimp assay indicated that functionalized ZnO NPs have improved cell imaging characteristics. It was discovered that the drug encapsulation efficiency was improved and the drug release was prolonged. The MTT assay revealed that they had a greater apoptotic effect on MCF-7 cells [92]. This sheds light on the intriguing potential of employing ZnO NPs as a promising agent in sustained release of bioactive anticancerous drugs.

### 5.2. ZnO NPs: Carrier of Conventional Chemotherapeutic Anticancerous Drugs in Sustained Drug Delivery

The cytotoxic medications doxorubicin (DOX) and paclitaxel (PTX) are employed in traditional therapy, but they have severe side effects since the processes they demonstrate are commonly shared by cancerous and normal cells. As a result, cytotoxic medications kill both types of cells [93]. ZnO NPs are considered as one of the most efficient theranostic agents for cancer treatment. Furthermore, the semiconductor characteristics with a large band-gap assist in the generation of ROS [94]. Dox-loaded ZnO NPs were evaluated on MCF-7 cell lines and results showed higher cytotoxicity and drug released in a controlled manner [93]. For efficient drug delivery, folic acid-functionalized polyethylene glycol coated ZO nanosheet (FA-PEG-ZnO NS) was synthesized and evaluated on MDA-MB-231 cells. An anticancer drug, doxorubicin, was loaded on to FA-ZnO NS. DOX-loaded FA-ZnO NS carried heat and drug expressively to cancer cells along with its targeted synergistic effect of chemo-photothermal therapy. Enhanced uptake and cytotoxicity were observed against breast cancer cells. Thus, the DOX-FA-ZnO NS system facilitates controlled drug release and targeted chemo-photothermal therapy in a single system [95]. Another study shows the potential of DOX-loaded ZnO NPs in breast and colon carcinoma (HT-29) and results affirmed that DOX-ZnO NPs are efficient in drug delivery to MCF-7 cells and HT-29 cells with minimum toxicity and high therapeutic efficacy [84].

ZnO NPs were also used to increase the drug retention time in targeted drug delivery. In a recent study, for prolonged drug release, DOX was loaded into a water dispersed ZnO-QD–chitosan–folate carrier. Encapsulation efficiency was discovered to be 75% as chitosan increases the stability of quantum dots (QDs) due to its hydrophilicity and cationic charge characteristics. The drug release response of the DOX-loaded ZnO-QD–chitosan–folate carrier was marked by a quick initial release followed by a regulated release. Results showed that the water dispersed ZnO QDs (2–4 nm) with long-term fluorescence stability could be exploited in the development of novel drug release carriers [96]. Likewise, for high transport of drugs, hollow nanocarriers are considered to be potential candidates since they provide substantial internal space in their core for drug loading [97,98]. Recently, a study was designed to establish a theranostic nanocarrier surface functionalized with folic acid for enhanced cellular uptake and unloading of paclitaxel drug in an acidic and malignant microenvironment. These ZnO nanocarriers showed a fluorescence reporting mechanism and drug release in a parallel manner. Thus, this “smart-targeting of chemotherapy” has the potential to improve the quality of life, recovery, and outcome of patients with breast cancer and possibly other organ tumors [99]. Moreover, the bacterial culture of *Rhodococcus pyridinivorans* NT2 was used to prepare anthraquinone ZnO NPs and cytotoxicity was checked against HT-29 cell lines. The MTT assay showed the ability of ZnO NPs to induce cytotoxicity in HT-29 cells in a dose-dependent manner. Thus, anthraquinone loaded ZnO NPs could be used as future candidates as anticancer drug delivery vehicles [100].

Ruthenium (Ru) has gained popularity as it exhibits an anticancer effect through its direct binding with DNA [101]. It tends to accumulate in neoplastic masses by using transferrin to invade tumors, leaving behind normal tissues and remaining in an inactive oxidation state, Ru (III) until it reaches the tumor site [102,103]. Surface-modified nanomaterials have the potential to deliver therapeutic compounds along with inhibition of cancer growth. Therefore, for efficient delivery of Ru pro-drug, ZnO-SiO_2_ core shell NPs were coated with polyethylamine and surface functionalized with cholic acid. ZnO-SiO_2_ NPs showed efficient Ru pro-drug delivery in cervical cancer treatment and tend to have a greater ability to successfully produce ROS in cancer cells. They were also found to be biocompatible and showed no acute toxicity. As a result, precise delivery of different therapeutic agents to their targeted areas was achieved, resulting in extremely efficient cancer therapy [104].

Likewise, microspheres have been created using hyaluronic acid (HA) as a gene delivery vehicle [105]. In cancer cells, the HA content rises [106], resulting in a less thick matrix, increased cell motility, and the capacity to invade healthy tissues. Because of its strong tumor selectivity and biocompatibility, HA could be employed to create tumor-targeting drug delivery vehicles for anticancer drugs like PTX. Thus, the HA coated poly butyl cyanoacrylate (PBCA) ZnO NPs were made by initiating radical polymerization of butyl cyanoarylate (BCA) in the presence of HA with cerium ions. A model anticancer agent, PTX, was encapsulated in negatively charged NPs with a 90 percent encapsulation rate. In vitro release showed that HA alteration reduced the first burst release in the first 10 h and provided a continuous release over the next 188 h. The hemolysis experiment and the MTT assay both showed that HA coating could greatly lower cytotoxicity. According to cellular uptake, HA-PBCA ZnO NPs were 9.5 times more effective than PBCA ZnO NPs in Sarcoma-180 (S-180) cells. As a result, HA-PBCA ZnO NPs may be an effective and safe vehicle for systemically administering hydrophobic anticancer drugs [107].

### 5.3. ZnO Nanocomposite: Carrier of Various Conventional/Bioactive Anticancerous Drugs in Sustained Drug Delivery

Nanocomposites have developed as viable options to overcome the limitations of various engineering materials. Nanocomposites are multiphasic materials having a diameter in the range of 10–100 nm in at least one phase [108]. Nanocomposites outperform standard microscale composites in terms of characteristics and may be synthesized using simple and low-cost processes. They can be prepared with enhanced physical, thermal, and other unique qualities [109]. Thus, they are widely employed to reduce the viability of cancers by enhanced drug delivery. As stated earlier, when bioactive compounds are conjugated to ZnO NPs, the results are superior to other NPs. To evaluate this, an MTT assay was carried out to investigate the anticancer properties of the synthesized ZnO conjugated L-asparaginase nanobiocomposite on the breast cancer cell line MCF-7. When MCF-7 cells were treated with L-asparaginase conjugated ZnO nanobiocomposite, their viability was reduced to 35.02%. This concludes the potential anticancerous activity of ZnO conjugated with L-asparaginase [110]. In another study, ZnO loaded quercetin nanocomposite were fabricated successfully by a cost-effective method. The synthesized ZnO nanocomposite was hexagonal in shape, having a diameter of 21–39 nm, which was used as a carrier. It was observed that the 210 µg/mg of quercetin was loaded onto ZnO nanocomposite in acidic conditions, which are typically more favorable for cancer cells to proliferate. Results showed that the ZnO nanocomposite was found to release quercetin at a faster rate than the targeted MCF-7 cancer cells [111].

Green synthesized nanomaterials have proven to be more efficient in targeted drug delivery as compared to conventionally synthesized nanomaterials. In a recent study, ZnO nanocomposites were synthesized using water extracts from the seaweed *Sargassum muticum* and hyaluronan biopolymer (HA). In this study, cell lines such as pancreatic (PANC-1), colonic adenocarcinoma (COLO205), and ovarian (CaOV-3) along with acute promyelocytic leukemia (HL-60) were treated with the HA/ZnO nanocomposite. After 72 h of exposure to a HA/ZnO nanocomposite, the MTT assay revealed that nanocomposites were most hazardous to HL-60 cells, inducing elevations in caspase 3/7 and inducing G2/M cell cycle arrest, while the normal human lung fibroblast (MRC-5) cell line remained unaffected. As a result of the findings, the HA/ZnO nanocomposite developed through green synthesis has a great potential as a cancer therapeutic agent [112].

Curcumin is widely known due to its ability to destroy cancer cells, but its poor water solubility and low bioavailability pose significant hurdles to its medicinal usage. Therefore, a study was designed to build a curcumin loaded poly methyl methacrylate (PMMA) and poly ethylene glycol (PEG) ZnO bio-nanocomposite containing insoluble curcumin and poorly soluble ZnO NPs to improve the bioavailability and ultimately the efficacy of curcumin. PEG and PMMA are suitable for usage in controlled-release polymer systems due to their demonstrated safety. The created nanocomposite was able to transport a substantial amount of curcumin while rapidly releasing its therapeutic payload at low pH, enhancing curcumin bioavailability and anticancer activity on gastric cancer (AGS) cells [79]. As a result, these bio-nanocomposites could be a potential anticancer medicine alternative in the near future.

**Table 1 cancers-13-04570-t001:** Role of various ZnO NPs-formulations in targeted/sustained delivery of anticancerous drugs.

ZnO NPs-Formulations	Morphology/Structure	Size	Drug Loaded	Cancer Cell Lines	IC_50_	Encapsulation Efficiency	Ref
Bioactive loaded ZnO NPs	Spherical	90 nm	Taxifolin	MCF-7	27 µg/mL	67.7%	[90]
	-	-	NRG, QE, CUR	A431	35, 25, 12 µg/mL	90.55%, 92.84%, 89.89%	[91]
	Oval	26 nm	CUR	MCF-7	-	85%	[92]
Conventional drug loaded ZnO NPs	Spherical	47.4 nm	DOX	MCF-7	-	85%	[93]
	-	160 nm	DOX	MBA-MB-231	-	89%	[95]
	Rod	55 nm	DOX	MCF-7, HT-29	0.125 µg/mL	-	[84]
	Monodispersed	2–4 nm	DOX	-	-	75%	[96]
	Spherical	125 nm	PTX	MCF-7, MDA-MB-231	14.02 nM, 11.84 nM	82%	[99]
	Spherical	100–120 nm	Anthraquinone	HT-29	-	79%	[100]
	Spherical	50 nm	Ruthenium	Hela	2 µg/mL	85.7%	[104]
	Monodispersed spheres	291–325 nm	PTX	S-180	-	90%	[107]
ZnO nanocomposites	Round	28–63 nm	-	MCF-7	-	-	[110]
	Hexagonal	21–39 nm	QE	MCF-7	0.01 µg/mL	-	[111]
	Hexagonal wurtzite	10.2 nm	Hyaluronan	PANC-1, CaOV-3, COLO205, HL-60	10.8 ± 0.3, 15.4 ± 1.2, 12.1 ± 0.9, 6.25 ± 0.5 µg/mL	-	[112]
	Hexagonal wurtzite	35 nm	CUR	AGS	0.01 µg/mL	92%	[79]

## 6. In Vitro and In Vivo Anticancerous Activity of ZnO NPs

Cancer is typically treated by conventional therapies such as chemotherapy, radiotherapy, and surgery. Although all these therapies seem to be very effective for killing cancer cells in theory, in fact, these nonselective therapies also introduce a lot of serious side effects [113,114]. Recently, nanotechnology-based nanosystems, with high biocompatibility, easily surface functionalization, cancer targeting, and drug delivery capacity, have demonstrated the potential to overcome these side effects. Among different nanomaterials, ZnO NPs are considered to be safe both in in vitro as well as in vivo. Owing to their highly biocompatible and biodegradable nature, ZnO NPs can be selected as potent nano-platforms for cancer treatment [115,116,117]. The use of ZnO NPs in cancer treatment has grabbed the interest of researchers and currently, a lot of studies have been published demonstrating the anticancerous activity of ZnO NPs [118,119,120]. Here, we have presented the anticancerous activity of ZnO NPs according to various types of cancers as summarized in Table 2.

### 6.1. Liver Cancer

Liver cancer is the only one of the top five most deadly malignancies that has an increase in occurrence annually [121]. Hepatitis B and C viruses, smoking, diabetes, obesity, and other dietary exposures are all risk factors associated with liver cancer. Liver cancer patients have a dismal prognosis [122]. Only 5% to 15% of people are candidates for surgical removal, which is only recommended for cancer patients in their early stages [123]. Long-term usage of chemotherapeutic medications, such as sorafenib, raises additional concerns of toxicity and/or pharmaceutical inefficacy. As a result, neither existing ablation treatments nor chemotherapies have been demonstrated to improve the outcomes of this severe disease appreciably [124]. In order to analyze the anticancerous efficiency of ZnO NPs, human hepatocellular carcinoma (HepG2) and normal rat cells (hepatocytes) were employed, and cytotoxicity was determined using the MTT assay. It was shown that concentrations up to 5 µg/mL did not result in a significant loss in viability of cells. However, concentrations of 10–15 µg/mL proved beneficial. HepG2 viability was reduced by 33%, whereas hepatocytes remained unaffected [40]. Similarly, at doses of 14–20 µg/mL for 12 h and 24 h, ZnO NPs showed a reduction in cell viability and apoptosis was induced in HepG2 cells [125]. In conclusion, ZnO NPs mediate ROS through the p53 pathway, which selectively triggers apoptosis in cancer cells.

Apoptosis and cell cycle regulation are intimately linked. As a result, lack of apoptotic processes leads to unregulated cell proliferation, which results in growth of cancer tissues and abnormalities [126]. The protein p53 has been found to be triggered in response to DNA damage induced by oxidative stress [127]. p53 also induces cell apoptosis and the activation of other genes involved in cell cycle checkpoint activation [128,129]. The Bax protein (a member of the Bcl2 family) is activated in response to numerous genotoxic stressors and interacts with the Bcl2 protein [130]. p53 stimulates Bax transcription directly [131]. The induction of Bax by activated p53 can counteract the anti-apoptotic effects of the Bcl2 protein. As a result, cells with a Bax deficiency are resistant to apoptosis-inducing stimuli [132,133]. In order to analyze this, an experiment was conducted to evaluate the cytotoxic and anticancerous activity of ZnO NPs on rat liver and spleen. Gene expression profiling using apoptotic markers, i.e., p53, BAX, and Bcl-2, was also carried out in this study. The results confirmed the upregulation of p53 and BAX genes, which were responsible for inducing apoptosis in human hepatoma cell lines (HepG2 and HUH7) [134].

Caspases are a type of cysteine protease that is thought to be involved in the apoptotic process. In mammalian cells, caspase-3 and caspase-9 have been identified as significant mediators of apoptosis. Their actions are regarded as adequate indicators of cytotoxic response [135]. ZnO NPs were exposed to a liver cancer cell line (Huh 7) and findings indicated that nano-ZnO stimulated autophagy, upregulated the expression of caspase 3 and p53 markers, and triggered apoptosis in liver cancer cells, hence limiting liver cancer cell growth and proliferation [136]. The mechanism suggested that caspase activation contributed to ZnO NPs triggered apoptotic death.

There are numerous ways of synthesizing NPs of desired shapes and sizes, including physical and chemical processes, but their limitations limit their application. Biological nanoparticle synthesis, on the other hand, is a novel and economically viable strategy in the field of “green chemistry” [137,138,139,140]. It is simple, safer, greener, and easier to scale up; it is also energy and cost efficient, and it can be accomplished under normal circumstances without the use of toxic substances [137,138,139]. Thus, the potential of ZnO NPs synthesized from *Eclipta prostrata* was tested on HepG2 cells, which showed that 100 mg/mL induced significant cytotoxic effects, whereas activation of caspase 3 and DNA fragmentation assays confirmed the apoptotic features of the cells [41]. Similar results were reported using a leaf extract of *Pandanus odorifer* [118]. ZnO NPs were also assessed in conjugation with *Geranium wallichianum* leaf extracts. The MTT assay was used to evaluate the cytotoxicity of the produced ZnO NPs against HepG2 and the findings showed that HepG2 cells exposed for 48 h to various dosages of ZnO NPs significantly reduced the metabolic activity of HepG2 cancer cells. Increases in ZnO NPs concentrations resulted in a steady decrease in metabolic activity. At 1000 µg/mL, the maximum inhibitory potential (71% mortality) was attained, and cytotoxicity potency decreased as concentration was decreased. Thus, the reduction in metabolic activity caused by ZnO NPs might be due to the anticancerous potential of these NPs [42]. Green synthesis can also be achieved by employing microorganisms such as bacteria, fungi, and algae. Thus, cytotoxic as well as anti-angiogenic effects of ZnO NPs synthesized from an algae, *Sargassum muticum*, were evaluated on HepG2 cells and results affirmed that concentrations of 2800 µg/mL at 72 h of exposure left only 4.5% of cells alive, thereby affecting a survival rate of 95%. This suggests that ZnO NPs synthesized from plant sources can be used as a supplemental drug in the treatment of cancer as it decreases angiogenesis and activates apoptosis [141].

### 6.2. Lung Cancer

Lung cancer continues to be the leading cause of cancer-related deaths in both men and women all over the world [142]. Cigarette smoking habits are strongly linked to lung cancer incidence and mortality. Lung cancer incidence and mortality are likely to rise in future decades as smoking rates peak, often first in males, then in women, before dropping following the implementation of comprehensive tobacco control initiatives [143,144,145]. In this regard, nanocomposites have proven favorable in several biomedical applications due to their unique electrical, mechanical, and optical properties [146,147,148]. In a study, a precipitation technique was employed to prepare a pectin-guar gum-zinc oxide (PEC-GG-ZnO) nanocomposite. This combination was employed as an immunomodulator to particularly attack cancer cells, which showed enhanced anticancerous effects by increasing concentrations from 25 to 200 µg/mL of the NPs. When compared to untreated human peripheral blood lymphocytes (PBL), pretreated PBL with nanocomposite showed increased cytotoxicity against lung cancer (A549) [149].

ZnO NPs have also attracted the attention of researchers due to their use for photothermal therapy (PTT) in treatment of cancer [95]. By generating heat and then irradiating it with near-infrared (NIR) light, cancer cells can be thermally abated. PTT may provide greater temporal-spatial selectivity and reduced invasiveness as compared to other established therapeutic approaches [150]. In a recent study, chemo-photothermal treatment of A549 cells was investigated utilizing a nano-sized composite of ZnO NPs and berberine (BER). A single hybrid nanocarrier containing BER and ZnO NPs could carry both entities to the tumor at the same time, resulting in boosting anticancer efficacies. Thus, BER-ZnO NPs were found to have improved anti-proliferative properties in A549 cells through increasing cytotoxicity. In vivo tests on rats revealed no evidence of renal or hepatotoxicity, respectively. As a result, BER-ZnO NPs could be safely used as an injectable formulation for chemo-photothermal therapy for lung cancer [151].

Plants have useful bioactive molecules such as alkaloids, phenols, flavonoids, saponins, and tannins that help them cope with stressful environments. NPs made from such promising plants are shown to have increased activity. As a result, A549 was tested using ZnO NPs synthesized from *Mangifera indica*. The influence of ZnO NPs on the viability of A549 cells was comparable to that of the standard medication used, cyclophosphamide. Increasing the concentration of NPs boosted their antioxidant activity and extremely stable ZnO NPs with substantial antioxidant and anticancer properties were found [152]. Likewise, ZnO NPs prepared from *Raphanus sativus* triggered cytotoxicity and showed significant results in the A549 cell line. Intense changes in morphology were observed after treatment, and the authors suggested that the high surface area to volume ratio of ZnO NPs may lead to enhanced cytotoxicity. Moreover, phyto-constituents present in the plant from which ZnO NPs was synthesized might also be responsible for the enhanced anticancer activity of these NPs [153]. Similarly, the A549 cancer cell line was treated with various dosages of ZnO NPs ranging from 1 to 100 µg/mL produced from *Pandanus odorifer* leaf extract. At dosages of 50 and 100 µg/mL, viability was reduced by 60–70 percent [118]. Thus, the biogenic approach to synthesizing ZnO NPs could prove useful in minimizing the use of toxic chemicals and elevating therapeutic efficiency.

### 6.3. Breast Cancer

The lack of equilibrium between proliferation and apoptosis represents a major hurdle for damaged cells to be cleared out through apoptosis. Activating apoptotic pathways in tumor-affected cells is an important part of cancer treatment [154]. The tumor suppressor p53 gene and the caspase enzyme aid in the regular examination of cells and their prevention of malignancy [155]. Zinc has a critical part in the stimulation of the apoptosis-inducing caspase-8 enzyme, as well as regulating the effect of p53, a tumor suppressor gene [156]. Caspase-9 is also an essential target of zinc [157], as it is a potent inducer of caspase-3 and other enzymes that cause nuclear membrane disintegration and cellular death. Hence, p53, Bax, and caspases are considered key apoptotic markers while targeting cancers. In order to evaluate this, MCF-7 was exposed to ZnO NPs, which showed inhibition of cancer cells in a dose-dependent manner. Apoptosis in MCF-7 was triggered since markers such as p53, Bax, JNK, and p21 were upregulated [119]. Similarly, activation of caspase 8 and the p53 pathway activated apoptosis in MCF-7 cells on exposure to ZnO NPs [158]. Likewise, biosynthesized ZnO NPs about the size of 40 nm and a spherical morphology induced apoptosis in a dose-dependent manner in the MCF-7 cell line [159]. Murine breast cancer cell lines, TUBO and MCF-7 were exposed to different doses of ZnO NPs and the expression of caspase 3 and caspase 8 genes was evaluated. At doses of 4 and 8 µg/mL, overexpression of caspase 3 was detected, whereas significant overexpression of caspase 8 was only observed at the dose of 8 µg/mL. These findings showed that ZnO NPs have the ability to induce apoptosis in both cancer cell types [160].

Recently, an extensive study on anti-tumor activity of ZnO NPs against various breast (MCF-7, MDA-MB-231) cancer cell lines was conducted. In this study, ZnO NPs (40 nm) induced cell selective toxicity and triggered apoptosis by enhancing BAX expression [161]. Likewise, two breast cancer cell lines, MCF-7 and T47D, were exposed to ZnO NPs, which showed dose-dependent inhibitory action and induced apoptosis. Results were confirmed via annexin V/PI staining, which showed no toxic effects on normal human embryonic kidney (HEK293) cells [162]. Thus, ZnO NPs can be efficiently employed as natural apoptosis inducers in human breast cancer cells.

Synthesis of ZnO NPs using a biogenic approach is a straightforward, practical, and competitive method in comparison to other NPs and delivers a high yield with a unique physical appearance [163]. Along with this, ZnO NPs have solely sparked a lot of interest recently, due to their wurzite structure, hexagonal phase, and n-type semiconductor [164]. Thus, ZnO NPs synthesized from *Gymnema Sylvestre* showed toxicity by inducing ROS, MMP damage and apoptotic morphological alteration. Activation of caspase and Bax was also observed in MCF-7 cell lines [165]. Cytotoxicity studies also showed *Pongamia pinnata* coated ZnO NPs at dosages greater than 50 µg/mL severely reduced the viability of breast cancer MCF-7 cells in a single treatment and also effectively inhibited the biofilm formation of Candida albicans [166]. As confirmed by MTT assay [118], synthesizing ZnO NPs using leaf extract of *Pandanus odorifer* proved beneficial in decreasing the viability of MCF-7 cell line. The MCF-7 cell line was also utilized to investigate anti-breast cancer cytotoxicity, and the findings demonstrate that ZnO NPs generated from root extract of Withania somnifera had significant cytotoxicity in a dose-dependent fashion [167]. ZnO NPs biosynthesized from *Knoxia sumatrensis* inhibited anti-proliferative activity on MCF-7 cell line [168]. In another study, ZnO nanorods were prepared using *Santalum album* and dose-dependent cytotoxicity was found against MCF-7 cells. Findings revealed that ZnO nanorods triggered apoptosis via an intrinsic mitochondrial route that was dependent on caspase activation [169]. Similarly, another study using ZnO nanorods synthesized from *Leea asiatica* plant extract against MCF-7 cancer cells reported similar findings [170].

In a recent report, ZnO NPs prepared from stem bark extracts of *Albizia lebbeck* were also evaluated for their anticancerous activity against the MCF-7 cell line. Cytotoxicity and induction of membrane blebs was found against strong (MDA-MB 231) and weak (MCF-7) metastatic breast cancer cell lines in a dose-dependent action and on comparison with 0.05 M and 0.01 M NPs, the 0.1 M ZnO NPs showed the best biological activity [171]. Similarly, biocompatible ZnO nanostructures were prepared from fruit extract of *Vateria indicia* (VI) with an average size of not more than 32 nm, and their cytotoxicity was assessed on human triple-negative breast cancer (MDA-MB468). ZnO-VI nanostructures suppressed human triple-negative breast cancer up to 91.18 ± 1.98% by promoting the generation of ROS via oxidative stress [172]. Likewise, ZnO NPs were synthesized from the leaves of *Cynara scolymus*. This plant is also well known to possess a wide range of compounds such as luteolin, glycosides, and apigenin [173]. Results showed that the synthesized ZnO NPs showed potential cytotoxicity against MCF-7 cells by causing apoptosis via enhanced oxidative stress and mitochondrial damage. They were also shown to reduce tumors associated with the liver, spleen, and kidney [174]. In another study, MCF-7 and VERO cell lines were exposed to spherical shaped ZnO NPs having a diameter of 65.9 nm. Anti-proliferative capability was found due to the induction of apoptosis and thus can be considered safe in development of treatment methods [175]. Hence, it can be concluded that the plant-mediated ZnO NPs could be the most efficient anticancerous agents due to their strong therapeutic potential and low systemic toxicity.

Apoptosis is a sort of controlled cell death that is one of the most popular ways for anticancer medications to generate cytotoxicity [148,176]. Several intracellular and extracellular factors have been found to activate an apoptotic cascade, including the generation of ROS, serum starvation, and ionizing radiation [177,178]. Mitochondria, the largest generator of ROS [177], are considered to be an active molecule in cell death pathways. DNA damage by ROS triggers p53. It is thus stated that ROS production by mitochondria is associated with apoptosis caused by p53 [179]. According to a study, PEG coated ZnO NPs are considered to be more stable than ZnO NPs alone. The anticancerous activity of both types of ZnO NPs was evaluated in this study and the results showed that the PEG coated ZnO NP-induced death in breast cancer cells by generating ROS and blocking the repair pathway at the same time, with low damage to normal cells as compared to naked ZnO NPs [180]. Similarly, a novel method of synthesizing ZnO NPs using egg albumin (EA) was employed and its anticancer performance on MCF-7 cells was evaluated. EA-ZnO NPs with a diameter of 20–60 nm induced ROS and upregulated p53, caspase 3, and caspase 9 while down-regulating Bcl-2 (an anti-apoptotic gene), resulting in cytotoxicity [181]. Thus, elevating apoptosis-inducing markers could be of potential importance in treating cancer.

### 6.4. Osteosarcoma

Osteosarcoma is a cancerous tumor that mostly affects the long bones, although it can also affect other bones in the body. It has a bimodal distribution with maxima in late adulthood and the second decade of life [182]. Osteosarcoma is characterized by nonspecific symptoms, the most prevalent of which are new-onset, strain-related pain lasting many months and sleep disruption due to pain [183]. Cancer deaths owing to malignant neoplasms of the bones and joints account for 8.9% of total cancer deaths in children and adolescents. The current treatment of osteosarcoma is complicated due to the high risk of local relapse in a large number of patients after chemotherapy [184]. Thus, there is a rapid need to develop anticancer agents with high specificity and low toxicity. Rehmanniae Radix (RR) is a nontoxic herbal remedy that is widely used in China [185]. Recently, a study was conducted in which ZnO NPs were synthesized from RR and their anticancerous activity was evaluated against the osteosarcoma cell line (MG-63). The proposed work showed that the supplementation of ZnO NPs on MG-63 cells triggered excessive generation of ROS, which led to an alteration in MMP and thus resulted in apoptosis. Induction of apoptosis and inhibition of metastasis against MG-63 cells by ZnO NPs might be inherited from RR, which is a rich source of bioactive compounds [186].

Alhagi maurorum is another important medicinal herb with a wide range of pharmaceutical uses [187]. Various components of A. maurorum and its products have been employed to cure various ailments, such as cancer, dropsy, asthma, bronchitis, and many others [188,189]. Thus, ZnO NPs were green synthesized using leaf extract of A. maurorum and were evaluated against several cell lines of osteosarcoma. Results showed that the ZnO NPs demonstrated anti-tumor osteosarcoma activity against HOS, MG-63, G-292, clone A141B1, Saos-2, and Hs 707 (A) cell lines in a dose-dependent manner, with minimal cytotoxicity to the normal cell line (HUVEC). By increasing intracellular release of dissolved zinc ions, ZnO NPs produced cytotoxicity in cancer cells, which was followed by increased ROS generation and cancer cell death via the apoptosis signaling pathway [190].

### 6.5. Colon Carcinoma

Colon cancer is also one of the most common type of cancer worldwide, and it ranks alongside lung, prostate, and breast cancer as one of the main causes of death [191]. Colon cancer is more likely in men and women over the age of 65 years. People in this age bracket are affected by 75% of all incidence tumors [192]. ZnO NPs, like other types of cancer treatments, also play its positive role in treating colon cancer. Plant-mediated ZnO NPs proved to be more efficient in treating colon cancer as compared to conventionally synthesized ZnO NPs due to their capping of phyto-chemicals. The activity of phyto-constituents against carcinogens could be mediated via the generation of ROS, which are implicated in phagocytosis, cell proliferation regulation, and intracellular signaling [193]. *Deverra tortusa* has gained attention in recent years due to the presence of several bioactive compounds such as flavonoids, glycosides, and terpenoids. The compounds are pharmacologically utilized in to treat asthma, hepatitis, and to regulate menstruation [194,195]. ZnO NPs prepared from aerial parts of *Deverra tortusa* were assessed against human colon adenocarcinoma (Caco-2) and A549 cell line. By inducing ROS, ZnO NPs displayed cytotoxicity against these two cell lines, whereas normal human lung fibroblast cell line (WI38) showed no significant cytotoxicity [193]. Similarly, reduced proliferation of Caco-2 cells was observed when exposed to ZnO NPs synthesized from *Arthrospira platensis*, whereas less toxicity was observed on the normal (WI38) cell line [196]. Likewise, selective toxicity was observed against HT-29 cell line on exposing ZnO NPs synthesized from rhizome extract of *Bergenia ciliata* [197].

Under normal circumstances, mitochondria create and release low quantities of ROS into the cytosol, which may act as signaling molecules for cell survival [198]. Intracellular NPs, on the other hand, cause cells to produce excessive amounts of ROS beyond the capacity of natural antioxidant defense mechanisms such as reductive GSH and antioxidant enzymes, resulting in cell death [199]. Compared to TiO_2_, SiO_2_, ZrO_2_, and carbon black nanomaterials, ZnO NPs cause much higher oxidative damage [199]. In a recent report, Caco-2 cells were exposed to silver and ZnO NPs to evaluate their respective cytotoxicity and it was found that ZnO NPs exert higher toxicity in comparison with silver NPs by elevating the levels of ROS [200]. Phyto-derived ZnO NPs from *Spondias pinnata* were evaluated on colon carcinoma (HCT-116) and chronic myelogenous leukemic (K562), along with normal lymphocytes/erythrocytes. Both phyto and chemically derived ZnO NPs were assessed in parallel and were hexagonal in shape and had an average size of 30 and 48.5 nm, respectively. Phyto-derived ZnO NPs showed cytotoxicity against both the cell lines, while the one derived chemically were toxic only on HCT-116 cells. Apoptosis was observed to be induced by oxidative stress, elevating levels of ROS and promoting loss of MMP. No deleterious effects were observed in cells exposed to phyto-derived ZnO NPs, whereas toxicity was induced on cells exposed to chemically derived ZnO NPs [201]. Likewise, biocompatible ZnO NPs synthesized from *Silybum marianum* showed cytotoxic activity against the hepato-cellular carcinoma (HepG2) human cells [202].

### 6.6. Cervical Cancer

Cervical cancer is the fourth most common female malignancy worldwide, and it poses a significant global health challenge [203]. Every year, more than 500,000 women are diagnosed with cervical cancer, and the illness kills over 300,000 people globally. In most cases, high-risk subtypes of the human papillomavirus (HPV) cause this disease. The general prognosis for women with metastatic or recurrent illness remains poor [204]. According to studies, NPs synthesized using a green approach could be applied to cure cancer sufferers in the near future with minimal toxicity. They are chosen for their inherent capacity to penetrate tissue and cells, as well as their propensity to interact with malignant cells [205,206]. A study published showed the reaction of exposing ZnO NPs to human cervical cancer (Hela) cell lines synthesized from Abutilon indicum L. Results showed the cytotoxicity of ZnO NPs by selectively killing cancer cells through the production of ROS via the p53 pathway [207]. Likewise, rhizome extract of Bergenia ciliata was also employed to prepare ZnO NPs which showed a remarkable selective cytotoxicity against the Hela cell line, reducing cell viability at significant rates [197].

Various types of ZnO nanostructures have also been used for treating cancer. In a recent study, novel ZnO nanostructures were tested for their anticancerous activity against Hela and normal HEK cell lines. Growth inhibition and cell death were boosted in a dose-dependent manner in the Hela cell line as compared to the normal HEK cell line. Cell death was mainly induced by the increase in the formation of micronuclei, and these nanostructures might interfere with the rejoining of DNA strand breaks. Thus, ZnO nanostructures exhibited potent cytotoxicity against Hela cell lines and to some extent to normal cells, at all tested concentrations [208].

### 6.7. Other Cancers

Human multiple myeloma (MM) is a malignant and incurable tumor of the B cell. Anemia, renal failure, immunodeficiency, infection, and hypercalcemia are the most common clinical symptoms of MM patients [209,210]. MM remains an incurable hematological cancer, and pharmacological therapy continues to face the obstacles of drug resistance [211] and adverse side effects [212]. As a result, one of the most significant tasks in myeloma therapy research is to overcome drug resistance in myeloma cells and to develop low-toxicity, high-efficiency medications. Hence, a study was designed to investigate the effects of ZnO NPs on RPMI8226, a human myeloid derived cell line. The apoptosis assay confirmed cell death in a time and dose-dependent manner. Additionally, ZnO NPs also increased ROS production and decreased ATP levels in MM cells. Elevated expression of apoptotic markers Cyt-C, Apaf-1, caspase 3, and caspase 9 were also observed, which induced cell death. In contrast, little cytotoxicity was shown in peripheral blood mononuclear cells (PBMCs) [213].

Nanostructures are widely used in cell and molecular biology, tissue engineering, clinical bio-analytical diagnostics, and therapies as markers and probes [214]. Because of their cytotoxic nature, quantum dots have the ability to slow the growth of cancer cells. In a study, the activity of ZnO QDs was evaluated against myoblast cancer cells (C2C12). Inhibition of the growth of cancer cells was observed in a dose-dependent action. Reverse transcription (RT) polymerase chain reaction analysis showed upregulation in caspase 3/7 genes, responsible for inducing death [215].

Tongue cancer is among the most frequent cancers in the oral maxillofacial region, and recurrence is common even after surgery. The potential of ZnO NPs on human tongue cancer cells (CAL 27) was investigated in a study. The viability of CAL 27 cells was reduced as ZnO NP concentrations were increased. By elevating intracellular ROS levels and reducing MMP, ZnO NPs triggered PINK1/Parkin-mediated mitophagy in a time-dependent manner [216].

Natural plant extracts and their bio constituents provide environmentally acceptable biosynthetic techniques for various metal and metal oxide NPs, allowing for a regulated synthesis with tunable form and size [217]. Therefore, ZnO NPs were prepared from the leaves of *Murraya keenigii* and their potential towards treating gastric cancer cells (MGC 803) were evaluated in vitro. Significant toxicity was induced against MGC 803 cell lines, extending the scope of ZnO NPs in biomedicine [218]. Efficient antibacterial and anticancer activities were observed by curcumin and ZnO NPs used either alone or in combination. Thus, curcumin loaded ZnO nanocomposites were fabricated and evaluated against bacterial strains and rhabdomyosarcoma (RD) cell lines. Spherical ZnO-curcumin NPs (SZNPs-CS) exhibited an excellent effect showing low toxicity against normal cells and higher toxicity against RD cell lines along with development of inhibition zones against tested bacterial strains [219]. *Marsdenia tenacissima*, a Chinese medicinal herb, has long been utilized as a clinical treatment for many types of cancers. ZnO NPs were prepared using extracts of *M. tenacissima* and were evaluated against the laryngeal cancer cell line Hep-2 in vitro. ZnO NPs were able to successfully generate ROS, alter MMP, and promote nuclear damage. RT-PCR analysis confirmed the upregulation of Bax, caspase 3, and caspase 9 while down-regulating Bcl-2. Thus, *M. tenacissima* mediated ZnO NPs could be a viable anticancer approach for treating a variety of cancers [117].

Ovarian cancer remains the most fatal of all gynecological cancers and the leading cause of cancer-related mortality in women. Oxidative and proteotoxic stress by ZnO NPs initiating apoptosis was evaluated on ovarian cancer cell lines and it was demonstrated that their viability considerably dropped with increased cytotoxicity as the size of NPs decreased [220]. In a study, the cytotoxic effect of ZnO NPs was investigated on human ovarian cancer cells (SKOV 3). According to the findings, ZnO NPs can cause significant cytotoxicity, apoptosis, and autophagy in human ovarian cells via producing ROS and oxidative stress [221].

The generation of ROS such as peroxide, hydroxyl radicals, and superoxide is strongly linked to cancer cell apoptosis [222,223]. As a result, gaining a better knowledge of the influence of ROS will lead to the development of viable strategies for developing novel and successful cancer medicines [224]. Oxidative stress caused by well-crystallized ZnO NPs was evaluated on Cloudman S91 melanoma cancer cells. Hexagonal ZnO NPs with an average size of 10 nm successfully induce the production of ROS causing apoptosis with varying doses of ZnO NPs [120]. Similarly, another research project looked into the effects of ZnO NPs on human gingival squamous cell cancer (GSCC). The inhibited growth of ZnO NPs was investigated against Ca9-22 and OECM-1 cell lines. Results predicted the selective anticancerous effects of ZnO NPs on GSCC by generating ROS and disrupting MMP, which leads to apoptosis through caspase-dependent pathways. Human normal keratinocytes (HaCaT cells) and gingival fibroblasts, on the other hand, appeared to be less affected by ZnO NPs [225].

Recently, ZnO nanostructures induced apoptosis in a dose-dependent manner in a human brain tumor (U87), confirmed via MTT assay [208]. Likewise, inorganic ZnO NPs were synthesized by a co-precipitation method and were evaluated in leukemia (K562) cells. ZnO NPs showed an inhibitory effect on K562 cell proliferation while being safe for lymphocyte normal cells [226]. In another study, varied murine cancer cell lines (CT-26, 4T1, CRL-1451, and WEHI-3) as well as a normal mouse fibroblast cell line (3T3) were also treated with different concentrations of ZnO NPs. The anti-proliferative effects were found and confirmed using the MTT assay. However, no toxic effect was shown on normal fibroblast cell lines [227]. This confirms the potential role of ZnO NPs as a key player in cancer therapies.

**Table 2 cancers-13-04570-t002:** Anticancerous activity of ZnO NPs on different types of cancer cell lines.

Method of Synthesis of ZnO NPs	Morphology/Structure	Size (nm)	Exposure Time	Cancer Type	Cell Line	IC_50_ Value	References
Chemical method	Polygonal	21 nm	24 h	Liver cancer	HepG2	10–15 µg/mL	[40]
-		30 nm	12, 24 h		HepG2	14.5 µg/mL	[125]
Biological method		20–40 nm	72 h, 24, and 48 h		HepG2 and HUH7	40 µg/mL, 17.5, and 15 µg/mL	[134]
Biological method	Spherical	90 nm	24 h		HepG2	-	[118]
Biological method	Multi-shaped	96–110 nm	24 h		HepG2	-	[41]
Biological method	Hexagonal	18 nm	48 h		HepG2	39.26 µg/mL	[42]
Biological method	-	-	48 h		HepG2	150 µg/mL	[141]
-	-	50 nm	24 and 48 h		HepG2	2.22 and 1.54 µg/mL	[228]
Chemical method	Hexagonal	50–70 nm	24 h	Lung cancer	A549	50 µg/mL	[149]
Biological method	Hexagonal wurtzite	60 nm	-		A549	-	[152]
Biological method	Spherical and hexagonal	209 nm	48 h		A549	40 µg/mL	[153]
Biological method	Spherical	90 nm	24 h		A549	-	[118]
Biological method	-	-	24 h	Breast cancer	MCF-7	121 µg/mL	[119]
Biological method	Spherical	90 nm	24 h		MCF-7	-	[118]
Chemical method	Round	10–15 nm	48 h		MCF-7	15.88 µg/mL	[158]
Biological method	Spherical	40 nm	24 h		MCF-7	40 µg/mL	[159]
Biological method	-	32.5 nm	24 h		MCF-7 and TUBO	40 and 33 µg/mL	[160]
Biological method	Spherical	40 nm	72 h		MCF-7, MDA-MB231	23.8 µg/mL, 41.354 µg/mL	[161]
Biological method	Spherical	81.1 nm	24 h		MCF-7	36 µg/mL	[165]
Biological method	Face centered cubic	30.2 nm	24 h		MCF-7	32.8 µg/mL	[166]
Biological method	Hexagonal wurtzite	32 nm	24 h		MCF-7	6.84 µg/mL	[167]
Biological method	-	-	24 h		MCF-7	58.87 µg/mL	[168]
Biological method	Rod	100 nm	48 h		MCF-7	10 µg/mL	[169]
Biological method	Multi-shaped	66.25 nm	24 h		MCF-7, MDA-MB 231	-	[171]
Biological method	Spherical	65.9 nm	-		MCF-7	65.31 µg/µL	[175]
Chemical method	Spherical and hexagonal	20–60 nm	24 h		MCF-7	100 µg/mL	[181]
Chemical method	Tetragonal	30–40 nm	48 h		MCF-7	33.06 µg/mL	[174]
Biological method	Hexagonal wurtzite	10–12 nm	24 h	Bone cancer	MG-63	-	[186]
Biological method	Spherical	27.92 nm	-		HOS, MG-63, G-292 clone A141B1, Saos-2, Hs 707(A)	234, 285, 327, 372, 341 µg/mL	[190]
Biological method	Hexagonal	15.22 nm	-	Colon cancer	Caco-2	50.81 µg/mL	[193]
Biological method	Spherical	30–55 nm	48 h		Caco-2	9.95 ppm	[196]
Chemical method	Hexagonal wurtzite	30 nm	48 h		HCT-116	60 µg/mL	[201]
Biological method	Flower shaped	30 nm	48 h		HT-29	124.3 µg/mL	[197]
Biological method	Spherical	50–500 nm	24 h	Cervical cancer	Hela	45.82 µg/mL	[207]
Biological method	Flower shaped	30 nm	48 h		Hela	101.7 µg/mL	[197]
Chemical method	Flower and hexagonal	20–30 nm	48 h		Hela	9.2–128 µg/mL	[208]
-	Spherical	30 nm	24 h	Human multiple myeloma	RPMI8226	33.83 µg/mL	[213]
Chemical method	Spherical	7–8 nm	24 h	Myoblast cancer	C2C12	-	[215]
-	Hexagonal	30 nm	24 h	Oral cancer	CAL 27	25 µg/mL	[216]
Biological method	Spherical	20 nm	24 h	Gastric cancer	MGC803	-	[218]
Chemical method	Spherical	40–100 nm	24 h	Rhabdomyosarcoma cell line		13 µg/mL	[219]
Biological method	Spherical	-	24 h	Laryngeal cancer	Hep-2	7.5 µg/mL	[117]
Chemical method	Flower and hexagonal	20–30 nm	48 h	Human brain tumor	U87	9.2–128 µg/mL	[208]
Chemical method	Spherical	10 nm	24 h	Ovarian cancer	SKOV3	-	[220]
-	Hexagonal wurtzite	20 nm	24 h		SKOV3	-	[221]
Chemical method	Hexagonal	10 nm	24 h	Melanoma cancer	Cloudman S91	-	[120]
Chemical method	-	-	24 h	Human gingival squamous cell carcinoma	Ca9-22 and OECM-1	17.4 µg/mL and 51.0 µg/mL	[225]
Biological method	Hexagonal wurtzite	10–15 nm	72 h	Murine cancer cells	4T1, CRL-1451, CT-26, WEHI-3B	21.7, 17.45, 11.75, 5.6 µg/mL	[227]

## 7. Mechanism Involved in Anticancerous Activity of ZnO NPs via Apoptosis Pathway

Formation of intracellular ROS is believed to be linked to the mitochondrial electron transport chain and it is believed that the anticancerous agents that invade cancer cells could also damage the electron transport chain, resulting in a massive release of ROS intracellularly [229,230]. Thus, elevating levels of ROS results in mitochondrial damage followed by a loss in balance of protein activities, ultimately leading to apoptosis [231]. ZnO NPs, therefore, present cytotoxicity in cancer cells due to the increased intracellular levels of dissolved zinc ions, followed by enhanced ROS production, causing cancer cell death via an apoptotic signaling pathway [232].

Figure 5 represents the possible mechanism involved in the anticancerous activity of ZnO NPs. Cancer cells can uptake the ZnO NPs by an endocytic route, and this entry route may differ depending upon the cell type involved. Confinement of ZnO NPs to vesicular structures, endosomes and then finally to lysosomes is due to the energy-dependent processes of NP uptake. Acidic pH of lysosomes may trigger the cytosolic release of ZnO NPs and Zn^2+^ ions, selectively inducing toxicity, thereby, causing apoptosis, necrosis, cell cycle arrest, and membrane damage by excessive ROS production. Entrance of Zn^2+^ ions may also take place through ion channels suppressing the activity of Bcl-2 markers (anti-apoptotic proteins), which in turn induces the expression of Bak/Bax (pro-apoptotic proteins) to promote permeabilization, followed by release of cytochrome c. Formation of a complex combining cytochrome c, along with apoptotic protease activating factor (Apaf-1) and pro-caspase 9, activates the Apoptosome. Activation of caspase 9 triggers caspase 3 and 7 gene expression and activity, which ultimately leads to apoptosis of cancer cells.

## 8. Conclusions and Future Prospects

Despite significant advancements in the field of diagnosis, sustained drug delivery and treatment of cancer, it continues to be the leading cause of death worldwide. No effective treatment of cancer has been developed so far and all traditional therapies and medications are constrained by negative effects. Thus, researchers are striving to develop novel ways in order to generate better diagnostic devices and treatments that tend to have greater specificity and efficiency along with lower toxicity. In this time of hopelessness, nanotechnology has sparked hope that life-threatening diseases such as cancer will be efficiently treated in the near future. Higher retention period, tunable morphology, and lower rate of agglomeration are the major features of nanomaterials that have made them emerge so rapidly in the field of cancer theranostic.

Among metal oxide NPs, ZnO NPs have powerful inhibitory effects against malignant cells due to their inherent toxicity, which they achieve via causing intracellular ROS generation and activating the apoptotic signaling pathway, making them a suitable choice for anticancer medicines. Furthermore, when used as drug carriers, ZnO NPs have been shown to increase the bioavailability of therapeutic medicines or biomolecules, resulting in improved therapy efficiency. This review comprehensively represents the therapeutic potential of ZnO NPs against numerous cancer cell lines both in vivo and in vitro along with their mechanisms of action. The effect of ZnO NPs differed among cancer types, demonstrating that, in addition to the unique features of ZnO NPs, the cellular response is crucial. ROS can thus be considered as a player molecule in triggering p53 pathways and caspase cascades, ultimately leading to apoptosis.

Despite the numerous applications, exposure to ZnO NPs poses a significant threat to human health and the ecosystem. Though ZnO NPs offer significant safety and biocompatibility, their unregulated and uncontrolled use may have intended consequences for the biological system. Considering the future potential of ZnO NPs, it is unavoidably necessary to have a better grasp of their toxicity. It is also believed that ZnO NPs would significantly advance medical progress and are predicted to make even more intriguing contributions in these domains. Although ZnO NPs have shown exceptional promise in the diagnosis and treatment of cancers, more in-depth and advanced analysis of ZnO NPs, detailed understanding of the cellular and molecular pathways, and clinical trials will be required in the future for better cancer theranostic.

## Figures and Tables

**Figure 1 cancers-13-04570-f001:**
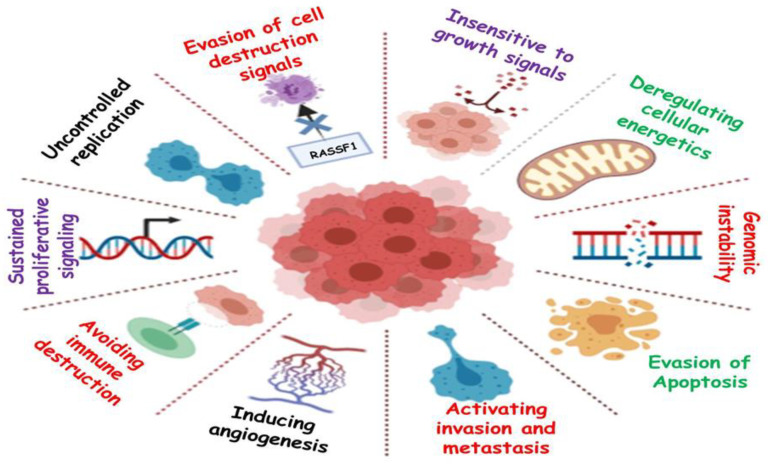
Graphical representation of the hallmarks of cancer.

**Figure 2 cancers-13-04570-f002:**
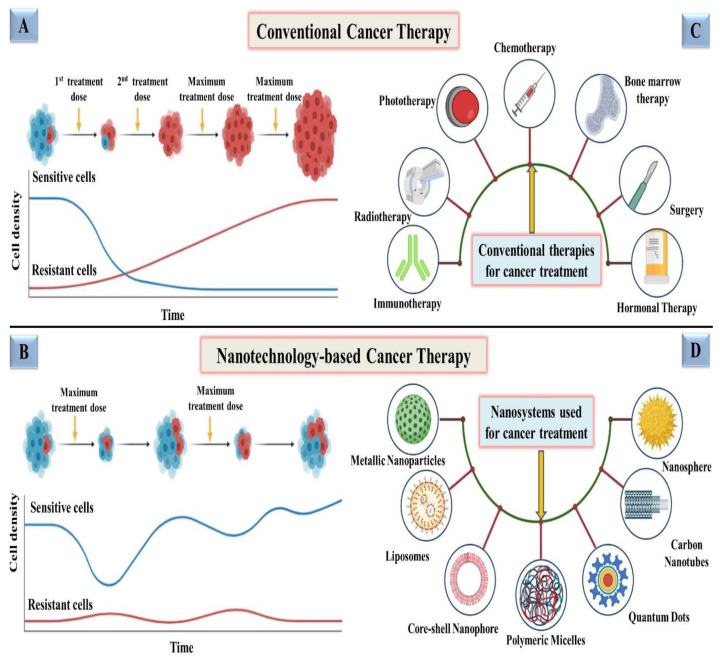
Schematic representation of a comparison between conventional vs. nanotechnology-based therapeutics for fighting cancer; (**A**) progression of resistance in cancer cells after exposure to subsequent doses of conventional therapies; (**B**) development of less resistance and more sensitivity in cancer cells after exposure to nanotechnology-based therapies even after maximum/higher doses; (**C**) representation of conventional therapies being employed for cancer treatment; (**D**) depiction of various nanotechnology-based nanosystems used for treating cancer.

**Figure 3 cancers-13-04570-f003:**
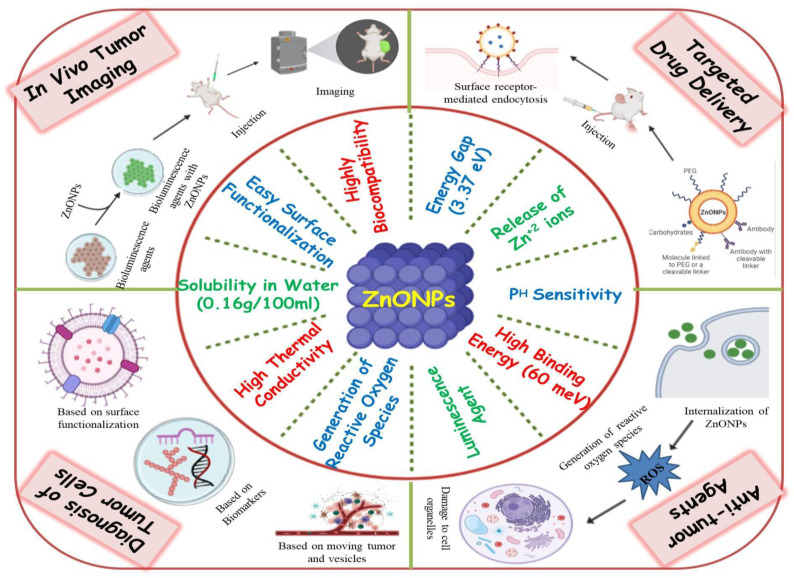
Graphical representation of the unique attributes of ZnO NPs and their various applications in fighting cancer.

**Figure 4 cancers-13-04570-f004:**
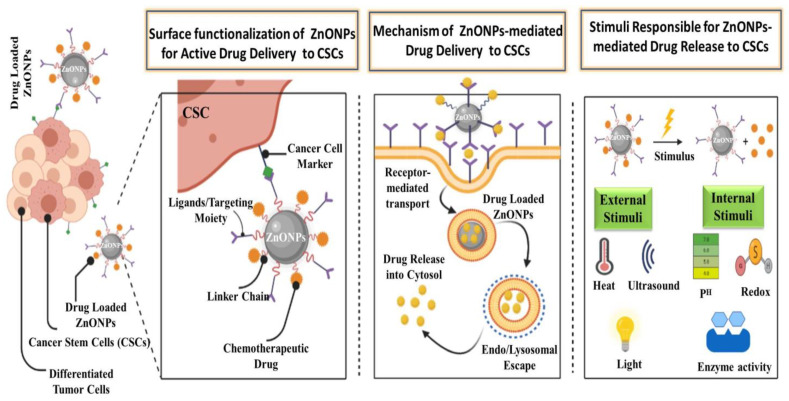
Graphical representation of the surface functionalization, mode of action, and various stimuli involved in targeted delivery of anticancerous drugs to cancer stem cells (CSCs) via ZnO NPs.

**Figure 5 cancers-13-04570-f005:**
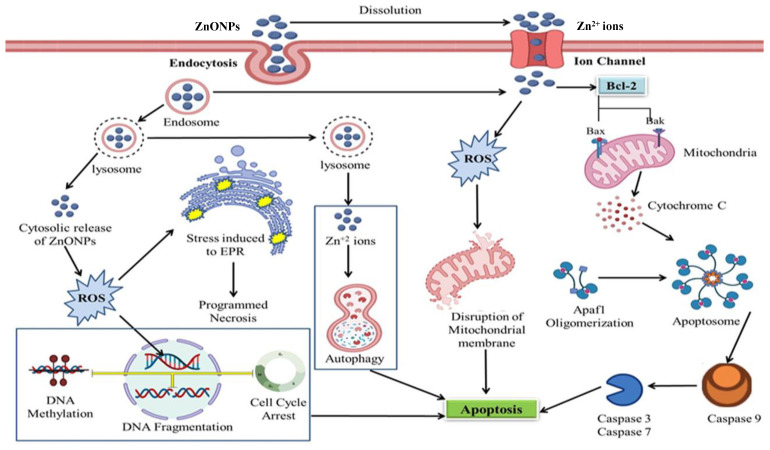
Possible mechanism involved in anticancerous activity of ZnO NPs.

## Data Availability

All data are included in present study.

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
