# Peer review of "Recent Advances in Zinc Oxide Nanoparticles (ZnO NPs) for Cancer Diagnosis, Target Drug Delivery, and Treatment"

_cancers, 2021, doi:10.3390/cancers13184570_

Round 1
Reviewer 1 Report
The manuscript entitled “Recent Advances in Zinc Oxide Nanoparticles (ZnO NPs) for Cancer Diagnosis, Target Drug Delivery and Treatment” by Sumaira Anjum et al, presents a review on the basic results and achievements in the advancing of synthesis and surface modification of Zinc Oxide Nanoparticles for Biomedical Applications and especially cancer disease.
This topic has been extensively developed in the literature and there are many interesting and major achievements in this field. This review is very detailed, and covers the major areas. However, there are many grammatical errors and surprising vocabulary which need to be corrected.
The tables and the figures are relevant and well-illustrated. One minor comment concerns the number of references. There are so many that it might be a good idea to reduce them somewhat.
Because of the uneven quality of the English the manuscript cannot be accepted in its present form. A suitably revised manuscript would be accepted.
Author Response
Reviewer-1
Comments: The manuscript entitled “Recent Advances in Zinc Oxide Nanoparticles (ZnO NPs) for Cancer Diagnosis, Target Drug Delivery and Treatment” by Sumaira Anjum et al, presents a review on the basic results and achievements in the advancing of synthesis and surface modification of Zinc Oxide Nanoparticles for Biomedical Applications and especially cancer disease. This topic has been extensively developed in the literature and there are many interesting and major achievements in this field. This review is very detailed, and covers the major areas.
AUTHORS: Thank you very much for your comments and suggestions that greatly help us to improve the quality of the present revised version of our MS. We do our best to answer to all your queries and hope this revised version will answer them. Revisions appear in track changes.
- There are many grammatical errors and surprising vocabulary which need to be corrected. Because of the uneven quality of the English the manuscript cannot be accepted in its present form. A suitably revised manuscript would be accepted.
AUTHORS: Thank you very much for your valuable suggestion. Grammatical errors and vocabulary has been revised thoroughly.
- The tables and the figures are relevant and well-illustrated. One minor comment concerns the number of references. There are so many that it might be a good idea to reduce them somewhat.
AUTHORS: Dear Sir, Thank you very much for appreciating our work. However, all the references included in the review article are relevant and up to date. Since no irrelevant reference is reported, so deleting references would affect the overall quality of the article.

Reviewer 2 Report
Anjum S et al., have submitted the review entitled “Recent Advances in Zinc Oxide Nanoparticles (ZnO NPs) for Cancer Diagnosis, Target Drug Delivery and Treatment”. The authors have provided a comprehensive review on ZnO NPs against cancer, including diagnostics.
The manuscript is well written and flawless. Each subtopic has been covered broadly and cited the ample of papers.
Figures and tables are well captured and will attract major cancer and biomaterial researchers.
This reviewer doesn’t have any questions or suggestions.
Hereby I endorse the manuscript for publication.
Author Response
Reviewer-2
Comments: The manuscript is well written and flawless. Each subtopic has been covered broadly and cited the ample of papers. Figures and tables are well captured and will attract major cancer and biomaterial researchers. This reviewer doesn’t have any questions or suggestions.
AUTHORS: Thank you very much for your positive comments.

Reviewer 3 Report
I do appreciate the paper as an exhaustive and well organized material, for a significant help of the scientific community. Research on topics related to the ZnO nanoparticles were extended in the last years, so this review represents a good tool in the choice of new R&I directions, once the cancer disease research needs new insights and advanced knowledge. The authors consulted multiple references in the recent years, including 2021. A special appreciation to be granted to the choice of sections subjects, as well as to suggestive corresponding graphics that are a considerable support to summarizing and/or deliver specific explanations. I suggest herein few minor corrections: - section 2 - carefully correlate the Fig.2 captions with the images A,B,C,D - it seems the text quoting the figure correlates with the images, but captions need to be corrected. - section 5 - Table 1 - second column, first row - according to info inserted in the following lines, I suggest to add word "Structure" - i.e. "Morphology/Structure" - section 6 - same observation as above - for Table 2 General observation for the sections where the safety use of ZnO NPs - I consider that it worth to include in the text several concrete criteria and related numbers that lead to this conclusion (quoting respective references and / or institutional reports - i.e. FDA). As an argument - the authors themselves declare in section 8 that further research on ZnO NPs toxicity is a good R&I direction.
Author Response
Reviewer-3
Comments: I do appreciate the paper as an exhaustive and well organized material, for a significant help of the scientific community. Research on topics related to the ZnO nanoparticles were extended in the last years, so this review represents a good tool in the choice of new R&I directions, once the cancer disease research needs new insights and advanced knowledge. The authors consulted multiple references in the recent years, including 2021. A special appreciation to be granted to the choice of sections subjects, as well as to suggestive corresponding graphics that are a considerable support to summarizing and/or deliver specific explanations. I suggest herein few minor corrections:
AUTHORS: Dear Sir/Madam, thank you very much for appreciation of our work. Thank you very much for your comments and suggestions that greatly help us to improve the quality of the present revised version of our MS. We do our best to answer to all your queries and hope this revised version will answer them. Revisions appear in track changes.
- Section 2 - carefully correlate the Fig.2 captions with the images A,B,C,D - it seems the text quoting the figure correlates with the images, but captions need to be corrected.
AUTHORS: Thank you for pointing out this mistake; the caption has been revised accordingly.
- Section 5 - Table 1 - second column, first row - according to info inserted in the following lines, I suggest to add word "Structure" - i.e. "Morphology/Structure" - section 6 - same observation as above for Table 2 .
AUTHORS: Thank you so much for your suggestion. The tables has been revised accordingly.
- General observation for the sections where the safety use of ZnO NPs - I consider that it worth to include in the text several concrete criteria and related numbers that lead to this conclusion (quoting respective references and / or institutional reports - i.e. FDA). As an argument - the authors themselves declare in section 8 that further research on ZnO NPs toxicity is a good R&I direction.
AUTHORS: Regarding the safety of ZnO NPs, more detail has been added in the revised manuscript. However, we cannot specify any fixed concentration/amount of ZnO NPs to be regarded as safe as it varies according to the conditions and environment.
